# Forget-Me-Not: Making Backdoor Hard to be Forgotten in Finetuning

## Abstract

Backdoor attacks are training time attacks that fool deep neural networks (DNNs) into misclassifying inputs containing a specific trigger, thus representing serious security risks. However, due to catastrophic forgetting, the backdoor inside the poisoned models can be gradually removed under advanced finetuning methods. It reduces the practicality of backdoor attacks since the pretrained models often undergo extra finetuning instead of being used as is, and the attacks gradually lose their robustness given various finetuning-based backdoor defenses. Particularly, recent work reveals that finetuning with a cyclical learning rate scheme can effectively mitigate almost all backdoor attacks. In this paper, we propose a new mechanism for developing backdoor models that significantly strengthens the durability of the generated backdoor. The key idea in this design is to coach the backdoor to become more robust by exposing it to a wider range of learning rates and clean-data-only training epochs. The backdoor models developed with our mechanism can bypass finetuning-based defenses and maintain the backdoor effect even under long and sophisticated finetuning processes. In addition, the backdoor in our backdoored models can persist even if the whole model is finetuned end-to-end with another task, causing a notable accuracy drop when the trigger is present. We demonstrate the effectiveness of our technique through empirical evaluation with various backdoor triggers on three popular benchmarks, including CIFAR-10, CelebA, and ImageNet-10.

## 1 Introduction

Deep learning has become essential in most modern AI systems thanks to its outstanding performance in almost every task. However, developing high-performing deep learning models (DNNs) is costly; it often requires a large set of training data, expensive and advanced hardware, and lengthy training time. Therefore, using pretrained models provided by third parties become a popular practice. Non-expert customers may acquire and deploy the pretrained models as is, while expert customers can finetune those models using their own data for the target tasks. Those open a loophole for backdoor attacks, an emerging security threat that has drawn increasing attention in recent years. Existing works in backdoor attack literature (Gu et al., 2017; Liu et al., 2018b; Salem et al., 2020; Barni et al., 2019; Liu et al., 2020) have demonstrated that by injecting a backdoor trigger, i.e., a specific pre-defined pattern such as a small square, to a small portion of the training data, the trained model will misclassify when facing inputs with the presence of this trigger. In contrast, on benign inputs, the poisoned model still behaves normally, which makes the attack hard to detect. The adversary can fool customers into deploying such a backdoor model in their systems, then use inputs with the backdoor trigger to manipulate the model's outputs for gaining illegal benefits or causing awful damages. As the research in this field has progressed, the attacks have become more powerful and sophisticated. More recent methods are capable of utilizing stealthier triggers that are visually imperceptible (Nguyen & Tran, 2021; Doan et al., 2021).

As a countermeasure against backdoor attacks, various backdoor defenses have been introduced. Existing defenses (Liu et al., 2018a; Wang et al., 2019; Gao et al., 2019; Tran et al., 2018; Doan et al., 2020; Wu & Wang, 2021; Zeng et al., 2021a) have made substantial efforts to detect and mitigate the effects of backdoor using various approaches, such as inverting triggers, splitting datasets, pruning the DNNs, or adversarial unlearning. While sophisticated defense techniques like unlearning and pruning effectively mitigate backdoors, they often come at the cost of sacrificing accuracy on

the original tasks. Conversely, fine-tuning-based defense offers a more balanced approach, partially restoring the model's utility. However, vanilla fine-tuning alone provides only modest backdoor mitigation (Liu et al., 2018a). Therefore, fine-tuning is often combined with other defense mechanisms to achieve superior defense performance (Liu et al., 2018a; Li et al., 2021).

Besides backdoor defenses, finetuning also appears in many practical AI systems as a post-processing step to revise the pretrained models to better fit the customer's need. Expert customers often do not use the pretrained model as is but finetune it using their own clean data. This process can serve various purposes: to adapt the model to the target data domain, to extend its capacity to handle edge cases, or even to completely change the target task.

Based on the observations above, we believe that a solid backdoor attack must be resilient to different finetuning techniques. It both helps the attack bypass finetuning-based defenses and extends its application to target a broader range of victim systems. However, a recent paper (Sha et al., 2022) argues that standard finetuning with a proper learning rate can mitigate backdoors from common backdoor attacks. It then proposes a finetuning process with a cyclical learning rate schedule to cover a wide range of learning rates and use it as an effective backdoor defense. This technique is called super-finetuning, and we empirically confirm its defense effectiveness. Besides, another paper (Zhu et al., 2023) also claims to figure out the weakness of the vanilla finetuning and boost the performance of existing defense methods based on looking for flat regions of the loss function.

In this paper, however, we will counter the aforementioned belief by designing backdoor attacks that can withstand all existing finetuning techniques, even the advanced ones like super- and SAM-finetuning. Specifically, we propose FMN (Forget-Me-Not), a novel mechanism to train backdoor models for the finetuning-resistant purpose. The key components include a cyclical learning rate schedule, which is directly inspired by super-finetuning, and a clean-backdoor interleave training procedure. First, the cyclic-learning-rate training coaches the produced backdoor to endure a wide range of learning rates, unlike the backdoor from common methods. Second, the clean-backdoor interleave training allows the backdoor model to practice mini-finetuning epochs, thereby strengthening the backdoor's permanence. By surviving these extreme training manners, the poisoned model will acquire a deep-rooted backdoor, which is hard to mitigate.

We run extensive experiments to verify the proposed method on three common benchmarks, including CIFAR-10, CelebA, and ImageNet10. Our training scheme can be effectively applied to a wide range of backdoor attacks, remarkably strengthening their durability while causing no harm to their efficacy and stealthiness. Finally, we show that our backdoor can long-last inside the deep models even after being finetuned for other tasks, causing a significant accuracy drop when testing on inputs with backdoor triggers. Note that while our backdoor training mechanism is general and can be applied to any task, we limit the paper's scope to handle the most-used image classification task.

## 2 BACKGROUND

### 2.1 THREAT MODEL

Backdoor attacks are techniques that poison a model to have a hidden destructive functionality. The backdoored model can perform genuinely on clean inputs but misbehave when a specific trigger pattern appears. The trigger pattern can be in any form, such as image patch, content blending, noise perturbation, or spatial warping. For image classification, backdoored models can return a predefined target label (normally incorrect) when the trigger is present, regardless of image content.

Backdoors can be injected into DNNs at different stages. In this work, we consider model poisoning during the training stage, which is commonly used by most backdoor attack and defense studies. In this scenario, the attacker has total control over the training process, and purposely trains the DNN with a backdoor. The poisoned network is then provided to the customer to deploy.

### 2.2 PREVIOUS BACKDOOR ATTACKS

BadNets (Gu et al., 2017), one of the earliest backdoor attacks, injects a fixed image patch as a trigger into a small portion of the data while flipping their labels to the target class. BadNets boosted remarkable successes on various datasets despite its simple scheme. After BadNets, many methods

have been proposed to change the attack mechanisms (Liu et al., 2018b; Yao et al., 2019; Rakin et al., 2020; Chen et al., 2021; Bober-Irizar et al., 2022) or trigger types.

Since we consider the most simple attack scenario in which the attacker has full control of the model development, the trigger design is more important. BadNets and early attacks often select the trigger pattern randomly. BadNets uses a random or hand-picked image patch as the backdoor trigger, while Chen et al. (2017) use a random image blending, Barni et al. (2019) use sinusoidal strips, and Nguyen & Tran (2021) employ a fixed image warping. In contrast, advanced attacks often optimize the trigger to achieve some desired properties. Liu et al. (2018b) compute the optimal patch-based backdoor pattern that can magnify a set of neuron activations for a fast and effective backdoor injection. Li et al. (2020) use the same reverse engineer technique but optimize a noise-like trigger pattern with $L_p$ regularization for invisible attacks. Nguyen & Tran (2020) train a generator to produce input-aware backdoor triggers. LIRA (Doan et al., 2021) jointly trains the trigger generation function and the target classifier for imperceptible and robust backdoor attacks. Recently, Narcissus (Zeng et al., 2022) optimizes an optimal trigger pattern using a surrogate clean classifier, aiming for efficient clean-label attacks.

### 2.3 BACKDOOR DEFENSE METHODS

The victim could be aware of or advised about the security threats at every stage of building and utilizing the model, thus they could apply defenses in all stages, ranging from data scanning (***data defense***) and model examination (***model defense***) to test-time monitoring after the model is deployed (***test-time defense***). Our paper focuses on finetuning-based defenses, a popular approach in model defense. Hence, in the following text, we first briefly introduce data and test-time defenses, then deep dive into model defense approaches, particularly finetuning-based algorithms.

**Data defense.** The defender aims to identify potentially poisoned data and then clean or remove them before using them to train their model. Defensive methods at this stage look at distinct characteristics of the data in their feature space (Chen et al., 2018) or their relationship with respect to the covariance matrix of the features (Tran et al., 2018), or detect unusual high-frequency traits (Zeng et al., 2021b).

**Test-time defense.** Defense at this stage aims to detect and remove malicious query samples. By observing the randomness in the model's prediction on perturbed inputs, STRIP Gao et al. (2019) identifies inputs with low entropy in the predicted classes as malicious. Neo (Udeshi et al., 2019), instead, locates the trigger region by searching for the smallest square-like block that is able to alter the network's prediction. More recently, Februus (Doan et al., 2020) utilizes GradCAM (Selvaraju et al., 2017) to identify abnormally small influential regions as potential triggers.

**Model defense.** This defense aims at identifying or mitigating the backdoor in a suspicious model using either a small set or zero benign data. We categorize the model defense methods into three main directions: reverse-engineering techniques, neuron pruning, and finetuning (FT).

Neural Cleanse (Wang et al., 2019) is a representative reverse-engineering defense. It reverse-engineers an input pattern for each output label, such that all samples stamped with the pattern are classified to the same label, then detects abnormally small patterns. Another reverse-engineering method, ABS (Liu et al., 2019), generates backdoor trigger candidates by scanning the neurons and then verifies these candidates on a small set of benign samples.

Fine-pruning (Liu et al., 2018a) pruned neurons that are dormant w.r.t clean inputs. ANP (Wu & Wang, 2021) further explored this idea and proposed using adversarial weight perturbation to amplify the differences between clean and backdoor-related neurons.

Finetuning a model with a small set of clean data resembles the process of training this model with only clean data. Based on the catastrophic forgetting phenomenon, this approach expects that the finetuned model will be free of the backdoors. However, Liu et al. (2018a) argued that although vanilla FT could provide some degree of backdoor purification, it is not strong enough to defend the model against advanced backdoor attacks. Hence, other techniques, such as pruning (Liu et al., 2018a) and distillation (Li et al., 2021), are often incorporated to fortify the defense. Another direction is to design more advanced finetuning methods by observing the loss landscape. Inspired by the study on learning rate (LR) of (Smith & Topin, 2019) to obtain super-convergence for fast training, Sha et al. (2022) proposed a method named super-finetuning. Super-finetuning consists of two

phases: the first phase cyclically finetunes the model from a minimal to a maximal learning rate, and the second phase repeats the same process as in the first phase but with a smaller maximal LR. Beside super-finetuning, FT-SAM, named after Sharpness-Awareness-Minimization (SAM) of (Foret et al., 2020), was recently introduced by (Zhu et al., 2023). FT-SAM searches for flat minimums; thus, its solution is more robust: a small perturbation of the parameters will not change the final prediction. Regardless of their advancement, we will prove that finetuning-based backdoor defenses cannot remove the backdoors produced by our novel backdoor training method. Note that there are some defenses only employ finetuning in a second stage of unlearning the backdoor behavior after a first stage of "reconstructing the trigger" Wang et al. (2019). Their effectiveness relies much on the first stage, thus out of our scope.

## 3 METHODOLOGY

### 3.1 PROBLEM OVERVIEW

In this section, we present the formulation of backdoor attacks in the classification setting. Given a domain $\mathcal{X}$ in which each item is categorized to one of the classes in $\mathcal{C} = \{0, 1, \ldots, m\}$, a user wants to obtain a classification function, says a mapping $f_\theta : \mathcal{X} \to \mathcal{C}$, which is parameterized by $\theta$. To achieve such objective, the classification function $f_\theta$ is trained on a dataset $\mathcal{S} = \big\{ (x_i, y_i) : x_i \in \mathcal{X}, y_i \in \mathcal{C}, i = 1, 2, \ldots, n \big\}$, where $y_i$ is the correct class of $x_i$. It is expected to produce the correct $y$ for any input $x \in \mathcal{X}$.

To attack the model, the attackers will build a backdoor model or feed poisoned data to users to build a faulty model, says $f_{\theta_{\mathrm{bd}}}$, which has the property

$$f_{\theta_{\mathrm{bd}}}(x) = y; \qquad f_{\theta_{\mathrm{bd}}}\big(\mathcal{B}(x)\big) = c(y), \tag{1}$$

for any input $x \in \mathcal{X}$ with the correct class as $y$, where $\mathcal{B}$ is a backdoor function learned by the attackers, and $c(y)$ is the target label they want the backdoor model to return. We consider the common *all-to-one* setting, in which the target label is fixed $c(y) = c \ \forall y$. It means that $f_{\theta_{\mathrm{bd}}}$ acts normally on ordinary input $x$ but will produce a wrong target class for poisoned input $\mathcal{B}(x)$. To achieve this, the parameters $\theta_{\mathrm{bd}}$, and the corresponding classification function $f_{\theta_{\mathrm{bd}}}$, are obtained by minimizing a loss function $\mathcal{L}$ trained over some poisoned dataset $\mathcal{S}'$

$$\theta_{\mathrm{bd}} = \arg\min_\theta \sum_{(x', y') \in \mathcal{S}'} \mathcal{L}\big(f_\theta(x'), y'\big) \tag{2}$$

where $\mathcal{L}$ is the common cross-entropy loss. The model $f_{\theta_{\mathrm{bd}}}$ is then given to the user, who will finetune the parameters $\theta_{\mathrm{bd}}$ to obtain $\theta_{\mathrm{ft}}$ for either backdoor defense or a different target task. For example, the objective function of the recent finetuning-based defense FT-SAM (Sha et al., 2022) is

$$\theta_{\mathrm{ft}} = \min_\theta \max_{\|\theta' - \theta\|_2 \leq \rho} \sum_{(x, y) \in \mathcal{S}_{\mathrm{sc}}} \mathcal{L}\big(f_{\theta'}(x), y\big),$$

where $\rho$ is a chosen constant, $\mathcal{S}_{\mathrm{sc}}$ denotes a small clean data set owned by the user, and the norm $\ell_2$ could be replaced by any other norm. Given an input $x$ and its true label $y$, the defender expects the finetuned $f_{\theta_{\mathrm{ft}}}$ to correctly classify $x$ and its corresponding poisoned input:

$$f_{\theta_{\mathrm{ft}}}(x) = f_{\theta_{\mathrm{ft}}}\big(\mathcal{B}(x)\big) = y.$$

In this work, we focus on designing a backdoor training process called FMN, such that existing finetuning procedures, such as the recent FT-SAM and super-finetuning, are not able to remove the backdoor. One advantage of FMN is that it is agnostic to the backdoor injection function, thereby increasing its generality and practicality.

### 3.2 COUNTERING FINETUNING

In this section, we discuss our design of a backdoor modeling procedure (Fig. 1) that can make the backdoor unforgettable after undergoing standard/advanced finetuning techniques.

**Training with cyclical learning rates:** The proposed method, inspired by super-finetuning, changes the learning rate cyclically during backdoor model training. Specifically, consider the backdoor

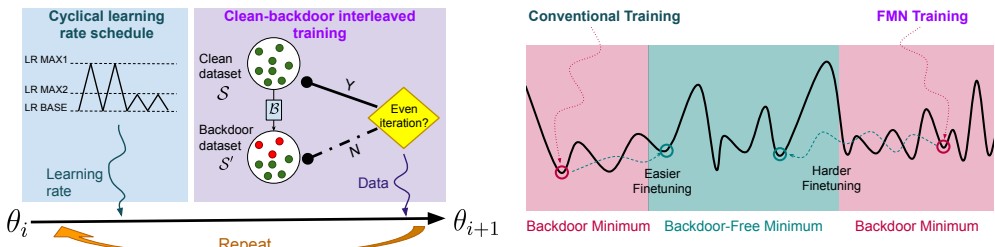

Figure 1: System overview (left) and the loss landscape discussion (right).

attack optimization problem as in Eq. 2 with initial model weights $\theta_0 = \theta_{\text{bd}}$. After $t$ updating steps, we obtain $\theta_{t+1}$ from $\theta_t$ as follows:

$$\theta_{t+1} = \theta_t - \epsilon \nabla_\theta \mathcal{L}_{\text{bd}}(\theta_t)$$

where $\epsilon$ is the learning rate, and $\mathcal{L}_{\text{bd}}(\theta_t)$ is the loss function's value on the poisoned data set. Instead of using the same $\epsilon$ as in the common backdoor training, we vary it using a designed schedule that consists of two phases. In the first phase, we define a maximum and a minimum learning rate. The learning is initialized with the minimum value and linearly increases to the maximum value in $n$ iterations, then linearly decreases back to the minimum value for another $n$ iterations. This cycle is repeated several times in this phase. The second phase is similar to the first one, except that we use a smaller maximum value. This novel design of training the poisoned model with cyclical learning rates exposes the backdoor to a wider range of learning rates during training, thus allowing it to be more robust to changes in the learning rate. Consequently, the backdoor becomes harder to remove, even when using advanced finetuning methods such as super-finetuning.

The main goal of finetuning defenses is to find an alternative local minimum that is free of the backdoor while preserving the model's utility. Cyclical backdoor training can also be viewed as searching for a region within the loss landscape where it is difficult for finetuning defenses to find these alternative local minima, as visually described in the right sub-figure of Fig. 1. In essence, this region primarily consists of local minima that are full of backdoors.

**Clean-backdoor interleaved training:** However, our experiments, which will be presented in detail in Sec. 4.5, reveal that relying solely on the cyclical learning rate in backdoor training is insufficient to achieve high attack effectiveness. Specifically, most finetuning defenses can significantly reduce the attack success rate of the backdoor trained with only cyclical learning, indicating that these methods could still jump to a local minimum where the backdoor's efficacy diminishes. Therefore, to further strengthen the backdoor's resistance against finetuning defenses, we propose a novel backdoor training strategy, called clean-backdoor interleaved training. This novel approach involves training the backdoor with the cyclical learning rate while additionally emulating the finetuning process during such training. This can be accomplished by training one epoch with only clean data immediately after one epoch of training on the poisoned data; this cycle is repeated until convergence. Intuitively, this approach allows the backdoor training process to form an even harder region in a hybrid loss landscape that is learned from the clean and conventional backdoor learning landscapes. The empirical analysis in Section 4 will show that the proposed approach is more resilient against fine-tuning defenses, confirming its ability to search for a difficult backdoor region for these defenses to break away from.

## 4 EXPERIMENTS

In this section, we examine the effectiveness of FMN against finetuning-based defense methods when applying this backdoor training mechanism to various backdoor attacks. We also verify the persistence of the backdoor produced by FMN undergoing transfer learning for a different downstream task. Finally, we confirm the utility of FMN by showing that it has no negative impact on the stealthiness of the original attack, illustrated by a case study with LIRA-FMN.

### 4.1 EXPERIMENTAL SETUP

We use three benchmark datasets, namely CIFAR-10 (Krizhevsky et al., 2009), ImageNet-10, and CelebA (Liu et al., 2015), for our experiments. To create the ImageNet-10 dataset, we randomly

Figure 2: Visualization of backdoor images on CIFAR-10 with different attack methods.

select 10 classes from ImageNet-1K (Deng et al., 2009). For CelebA, we follow the recommended configuration from Salem et al. (2020) to choose the three most balanced attributes, namely Heavy Makeup, Mouth Slightly Open, and Smiling, and concatenate them to form eight compound classes. To construct $f$, we utilize the Pre-activation ResNet-18 He et al. (2016) for CIFAR-10 and ResNet-18 for both ImageNet-10 and CelebA. We leverage the SGD optimizer for training the classifier $f$. For the cyclical learning rate, we choose $3e^{-4}$ for the minimum learning rate, 0.1 and 0.001 for the first and the second maximum learning rate, respectively, as recommended in Sha et al. (2022).

## 4.2 ATTACK EXPERIMENTS

We consider 7 attack methods for evaluation, which are representatives of different approaches, including BadNets (random patch-based trigger), Blended trigger attack (random blending-based trigger), $L_p$ regularization invisible attack, Trojaning attack (optimized trigger pattern), Input-Aware (image-dependent trigger), LIRA (imperceptible, optimized trigger), and Narcissus (clean-label attack with an optimized trigger). We showcase the backdoor images of these attack methods in Figure 2. For all these methods, we poison 5% of the training data and set the target label to 0.

**BadNets (Gu et al., 2017):** This is the first and most basic backdoor attack, in which the backdoor pattern is just a hand-picked image patch. We use a random color square at the top-left corner with $^1/_8$ the image size as the backdoor trigger on all datasets .

**Blended trigger attack (Blend) (Chen et al., 2017):** This attack chooses an outer image, e.g., a Hello Kitty image, and blends it with clean samples to create backdoor data.

**$L_p$ regularization invisible attack (Li et al., 2020):** This attack generates the backdoor trigger through an optimization problem while solving an $L_p$-norm constraint to guarantee the imperceptibility of the trigger. We choose $p = 0$ ($L_0$ inv) and $p = 2$ ($L_2$ inv) for our experiments. Since the code to generate the backdoor trigger has not been released, and only the triggers for CIFAR-10 are provided, we skip experiments with this attack for CelebA and ImageNet-10.

**Trojaning attack (Liu et al., 2018b):** This method looks for neurons and triggers that have strong connections so that when the trigger is present, the corresponding neuron is activated, which consequently activates the backdoor. The model is trained partly with generated data and the trigger is obtained by reverse engineering techniques.

**Input-aware (Nguyen & Tran, 2020):** used an encoder-decoder structure to create a trigger from the image, and then embed this trigger into the image. This approach together with the cross-trigger mode guarantees that a trigger generated for an image is unique to that image.

**LIRA (Doan et al., 2021):** This attack can produce visually imperceptible triggers and achieve high attack success rates by first simultaneously learning the backdoor injection function $\mathcal{B}$ and the classifier $f$, then finetuning $f$ on clean and poisoned data generated by $\mathcal{B}$.

**Narcissus (Zeng et al., 2022):** proposed to look for an optimal pattern among possible trigger patterns by minimizing a surrogate loss function.

We found that BadNets fails to converge when using the clean-backdoor interleaved training in FMN. In contrast, all other attacks can converge successfully. We conjecture that BadNets's random trigger patch is too prominent and different from image content, making its backdoor easy to learn but also easy to forget in finetuning. Hence, BadNets's backdoor cannot survive the clean-data-only training epochs. In contrast, Blend's trigger pattern is stronger by blending nicely with the image, while the other methods use optimization to learn optimal triggers that can survive the harsh training process of FMN. From now on, we will exclude BadNets and only consider the remaining attacks.

Table 1: **Performance of different backdoor attacks trained with FMN.** For each attack, we report the BA (%) in teal and ASR (%) in purple. The asterisk (*) denotes that the attack is trained with FMN. We skip experiments with $L_p$ inv on CelebA and ImageNet-10 since both the trigger generation code and trigger patterns for these datasets are not available.

| Dataset | Blend* | $L_0$ inv* | $L_2$ inv* | Trojaning* | Input-aware* | LIRA* | Narcisuss* |
|---|---|---|---|---|---|---|---|
| CIFAR-10 | 93.02/100 | 93.69/99.84 | 93. 43/100 | 94.00/99.98 | 94.23/99.81 | 94.26/100 | 94.08/99.68 |
| CelebA | 78.54/99.86 | - | - | 78.65/99.92 | 79.09/98.45 | 78.96/99.98 | 78.52/98.85 |
| ImageNet-10 | 84.86/91.54 | - | - | 85.61/92.72 | 86.59/90.42 | 88.60/95.98 | 85.23/94.26 |

We show the experimental results of training the attack methods with FMN in Table 1. For all attacks, we poison 5% of the training data. Every method can achieve a high attack success rate (ASR) and benign accuracy (BA), similar to backdoor models trained with the conventional mechanism.

## 4.3 FINETUNING-BASED DEFENSE EXPERIMENTS

Next, we evaluate the attacks developed with FMN in the previous section against prominent finetuning-based defense methods, including standard finetuning, super-finetuning (Sha et al., 2022), FT-SAM (Zhu et al., 2023), and NAD (Li et al., 2021).

**Standard finetuning (FT):** Finetuning is a technique that allows a pre-trained model to learn from new data, which was originally proposed in transfer learning to adapt a model to a new task using information learned in the pre-training phase. In the context of backdoor defense, finetuning a poisoned model on clean data is supposed to mitigate the backdoor due to catastrophic forgetting gradually. In our experiments, we adopt whole-model finetuning and use the same learning rate throughout the finetuning procedure. We choose two different learning-rate values, 0.01 and 0.05, to evaluate the influence of the learning rate choice on the defense performance.

**Super-finetuning (super-FT):** Based on the observation that large learning rates often help the model forget the backdoor trigger while small learning rates can maintain the model's utility, Sha et al. (2022) proposed a novel finetuning scheme, super-finetuning, that combines the two learning rates with a cyclical scheduler. The method includes two phases. In the first phase, the learning rate is repeatedly increased linearly from the base learning rate (LR BASE) to the first maximum learning rate (LR MAX1) and then dropped back to LR BASE. In the second phase, the learning rate varies with the same schedule but with a smaller maximum learning rate (LR MAX2). Following the original work, we set LR BASE to $3e^{-4}$, LR MAX1 to 0.1, and LR MAX2 to 0.001.

**FT-SAM:** Observing that backdoor-related neurons often have larger norms, Zhu et al. (2023) proposed to incorporate finetuning with Sharpness-aware Minimization (SAM) to shrink the norms of these neurons. Since the source code of FT-SAM has not been released, we implement FT-SAM in our experiments by simply replacing the optimizer in vanilla FT with SAM. We set SAM's perturbation radius $\rho$ to 2, as recommended in the original work.

**Neural Attention Distillation (NAD):** Arguing that finetuning alone is not a sufficient defense against backdoor attacks, NAD adopts a knowledge distillation (Hinton et al., 2015) technique for backdoor mitigation. It first obtains a teacher model by finetuning the backdoored model, then utilizes this teacher model to guide the finetuning process of the backdoored student model.

We run FT, super-FT, and FT-SAM for 100 epochs. For NAD, we first fine-tune the backdoor model for 20 epochs, then use it in conjunction with the student model through the NAD process and train for another 20 epochs. Following their original settings, defenses are allowed to access 5% of clean data, except for super-FT. As Sha et al. (2022) claim that super-FT is less effective when under 10% of clean data is available, we allow it to run with 20% of clean data for the best defense performance.

We show the experimental results on CIFAR-10 in Table 2. Additional results for CelebA and ImageNet-10 can be found in Appendix A.5. We also run these attacks trained with their original settings against these FT-based defenses for comparison. We choose 50% ASR as the threshold to determine whether an attack is successful since this level of accuracy indicates that backdoor samples are more likely to be misclassified as the target label. In the case of conventional backdoor training, finetuning with a low learning rate (0.01) can better preserve the model's utility but cannot effectively mitigate the backdoor, while finetuning with a large learning rate (0.05) has a contrastive effect. Combining both types of learning rates, Super-FT can mitigate all the conventional backdoor

Table 2: **Performance of conventional backdoor training and FMN against finetuning-based defenses on CIFAR-10.** For each attack, we report the BA (%) in teal and ASR (%) in purple. The asterisk (*) denotes that the attack is trained with FMN. The ASRs below $50\%$ are underlined.

| Attack | No defense | FT (lr = 0.01) | FT (lr = 0.05) | Super-FT | FT-SAM | NAD |
|---|---|---|---|---|---|---|
| Blend | 93.75/100 | 92.10/54.93 | 76.99/2.98 | 90.97/18.58 | 89.14/28.34 | 89.96/22.24 |
| $L_0$ inv | 92.98/100 | 90.96/69.65 | 74.53/1.29 | 91.08/17.26 | 88.65/25.14 | 90.66/6.98 |
| $L_2$ inv | 93.24/100 | 91.68/71.12 | 77.19/1.64 | 90.06/27.45 | 89.26/34.42 | 91.22/2.68 |
| Trojaning | 93.50/100 | 90.65/77.26 | 77.22/13.94 | 91.22/24.45 | 90.62/28.69 | 90.67/80.12 |
| Input-aware | 94.15/99.30 | 91.93/72.23 | 76.45/8.96 | 91.20/16.35 | 90.50/20.16 | 89.98/20.18 |
| LIRA | 94.42/100 | 90.25/88.63 | 77.08/10.24 | 90.98/28.75 | 91.04/32.26 | 90.17/78.65 |
| Narcisuss | 93.52/99.80 | 90.43/80.24 | 76.64/9.85 | 90.65/28.50 | 90.80/30.25 | 90.28/49.76 |
| Blend* | 93.02/100 | 92.52/99.66 | 82.57/71.20 | 91.99/86.22 | 92.05/98.54 | 92.46/54.64 |
| $L_0$ inv* | 93.69/99.84 | 91.18/98.77 | 80.68/65.79 | 91.74/86.65 | 92.69/99.84 | 91.65/49.56 |
| $L_2$ inv* | 93.43/100 | 91.22/99.87 | 85.08/71.26 | 92.53/87.90 | 93.02/98.52 | 92.56/70.74 |
| Trojaning* | 94.00/99.98 | 92.35/99.61 | 84.29/80.21 | 92.38/88.23 | 92.31/99.79 | 91.73/96.76 |
| Input-aware* | 94.23/99.81 | 93.02/95.25 | 79.65/68.49 | 91.29/80.46 | 91.33/94.24 | 90.26/76.49 |
| LIRA* | 94.26/100 | 92.26/96.49 | 81.75/82.30 | 91.45/88.69 | 92.65/99.68 | 92.04/90.22 |
| Narcisuss* | 94.08/99.68 | 92.56/96.35 | 79.91/74.82 | 90.55/86.96 | 90.42/96.76 | 90.34/79.64 |

models. Both FT-SAM and NAD can maintain high clean accuracy and provide a degree of backdoor mitigation. On the other hand, FMN can significantly improve the backdoor's durability in all the aforementioned defenses: FT (lr = 0.01) and FT-SAM have negligible effect on the ASRs, and all attacks' ASRs remain higher than 80% in super-FT. While FT (lr = 0.05) and NAD can mitigate the backdoor to some extent, most attacks still can achieve at least 70% ASR.

## 4.4 Transfer learning attack

In practice, users might not directly deploy the pre-trained backdoor model they receive from the attackers. Instead, they can finetune the poisoned model using their own clean data for specific purposes. Our attack design aims to make the backdoor survive even the most extreme finetuning case, where the user finetuned our backdoor model for a completely different downstream task. We set up these transfer-learning experiments by finetuning the ImageNet-10 backdoor models using data from 10 other classes of ImageNet-1k, which we denote as ImageNet-10-FT. Instead of re-training only the fully connected layers while keeping the convolutional layers intact, as in the transfer learning attack experiments in BadNets paper (Gu et al., 2017), we finetune the whole pre-trained model for a more challenging scenario. We test the finetuned backdoor model with clean and poisoned samples of ImageNet-10-FT. For reference, we also build a clean baseline by training a clean ImageNet-10 classification model and finetuning it with ImageNet-10-FT. A backdoor attack has a long-lasting effect if its finetuned model can achieve high accuracy on clean inputs of ImageNet-10-FT, similar to the clean baseline, while having low accuracy on the corresponding backdoor inputs.

We provide in Table 3 the results of this transfer learning scenario with two types of trigger, Blend and Trojaning. The conventional backdoors only cause a small accuracy drop (less than 8%) when their finetuned models are tested on backdoor inputs. In contrast, FMN causes significantly higher drops. It indicates that FMN enhances the backdoor's durability and consequently makes the backdoor effect remain even when the poisoned model is finetuned with a different downstream task.

## 4.5 Ablation studies

**Role of cyclical learning rate.** To validate the importance of the cyclical learning rate in our design, we contrast the proposed method with a baseline that only relies on the clean-backdoor interleaved training. As shown in Table 4, that baseline is quite underwhelming compared to FMN, confirming that including the cyclical learning rate in the training procedure can further strengthen the backdoor's resistance to fine-tuning.

**Role of clean-backdoor interleaved training.** To counter super-FT, a more straightforward method is to train the backdoor model with the cyclical learning rate. However, empirical results show that this approach is not enough to help the backdoor survive advanced finetuning defenses. As shown in Table 4, while this naive approach can provide certain improvements to the attacks' performance

Table 3: **Transfer learning experiments.** We provide the average accuracy (%) of finetuned clean model, conventional backdoor model, and FMN model with clean inputs in teal and backdoor inputs in purple. The difference in accuracy on backdoor inputs between each finetuned backdoor model and its clean baseline is provided as subscripted text.

| Attack | Clean Model | Conventional Backdoor | FMN |
|---|---|---|---|
| Blend | 82.36/78.29 | 80.56/74.68 $_{3.61\downarrow}$ | 80.86/54.42 $_{23.87\downarrow}$ |
| Trojaning | 83.14/79.98 | 81.88/72.32 $_{7.66\downarrow}$ | 82.04/48.95 $_{31.03\downarrow}$ |

Table 4: **Ablation studies.** We examine performance of training the attacks with clean-backdoor interleaved only and with cyclical learning rate only. We report the BA (%) in teal and ASR (%) in purple. The ASRs below $50\%$ are underlined. These experiments are conducted on CIFAR-10.

| Attack | Clean-backdoor interleaved training only | | | Cyclical learning rate only | | |
|---|---|---|---|---|---|---|
| | No defense | Super-FT | FT-SAM | No defense | Super-FT | FT-SAM |
| Blend | 93.74/99.97 | 90.55/54.86 | 88.46/49.72 | 93.68/100 | 91.21/55.28 | 88.65/64.85 |
| $L_0$ inv | 93.07/99.96 | 90.23/44.68 | 88.32/39.52 | 92.96/100 | 91.13/43.26 | 89.05/43.66 |
| $L_2$ inv | 93.35/100 | 91.24/64.27 | 90.15/47.82 | 93.20/100 | 91.26/64.62 | 90.14/54.71 |
| Trojaning | 93.75/100 | 91.35/39.86 | 90.75/47.92 | 93.59/100 | 90.85/68.64 | 90.12/62.05 |

against finetuning-based defense, it is constantly outperformed by FMN. As the results indicate, clean-backdoor interleaved training can indeed strengthen the backdoor's durability.

In summary, both the uses of cyclical learning rate and clean-backdoor interleaved training are essential to maintain strong attack success rates against finetuning defenses. While the cyclical learning rate allows backdoor training to search for a robust backdoor region in the loss landscape, clean-backdoor interleaved training further selects a more resilient one against finetuning defenses.

### 4.6 DOES FMN MAINTAIN THE GOOD PROPERTIES OF THE ORIGINAL ATTACK?

FMN is a simple training mechanism that can plug into most existing backdoor attacks to remarkably boost their durability against any finetuning technique. A natural question is whether FMN has any adverse effect on the produced backdoor. As shown in previous sections, FMN can maintain the attack's high efficiency. In this section, we will examine if FMN can keep the stealthiness of the original attack when being tested under other backdoor defenses. We pick LIRA as the representative attack, which has the conventional backdoor model bypassing a wide range of backdoor defenses, and test if its FMN model can go past the same set of defenses on CIFAR-10. We use four non-finetuning backdoor defenses, including Neural Cleanse (Wang et al., 2019), Fine-Pruning (Liu et al., 2018a), Adversarial neuron pruning (ANP) (Wu & Wang, 2021), and STRIP Gao et al. (2019). The LIRA model trained with FMN passes all the tests, as provided in detail in Appendix A.6. It confirms that FMN has no negative impact and can be safely used to strengthen backdoor methods.

## 5 CONCLUSIONS AND FUTURE WORKS

In this paper, we propose a novel training mechanism for backdoor models that can significantly strengthen their backdoor durability when these models undergo different finetuning processes. Our design includes the cyclical learning rate and the clean-backdoor interleaved training process. Extensive empirical experiments show that our design outperforms conventional backdoor training.

Our research highlights the potential risks of relying on third-party pre-trained models and underscores the importance of fostering trust between users and model providers. To defend against our attack, users should only use pre-trained models from trusted providers or actively participate in the training process. We also urge the research community to delve further into this area to develop more robust safeguards.

One limitation of this work is that it requires the attackers to have full access to the backdoor model training. While this setting is commonly studied in the literature, it might not always be true in practice. Extending our design to craft an attack that can survive finetuning in black-box settings would be an exciting future research direction.

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
