# OpenReview forum: "Forget-Me-Not: Making Backdoor Hard to be Forgotten in Fine-tuning"
_ICLR.cc/2024/Conference — Submitted to ICLR 2024_

### Official Review · Reviewer_EHtc · 2023-10-21

**Soundness:** 2 fair
**Presentation:** 2 fair
**Contribution:** 2 fair
**Rating:** 3
**Confidence:** 5

**Summary:**

This paper studies how to maintain stable attack performance of inserted backdoor triggers during robust tuning process. Specific to current superFT method, the authors also adopt a cyclical learning rate scheme during inserting backdoor triggers.

**Strengths:**

1. The studied problem is important. How to construct more robustly inserted backdoor triggers is interesting.

**Weaknesses:**

Limitations and lacking of ture explanations.
1. Limitations:
- The proposed method is limited: The proposed attack method is specific to superFT which adopt a cyclical learning rate scheme. It is not clear whether other defense methods are equally effective, such as ANP [1]. I believe adopting a cyclical learning rate scheme is not a proper choice for inserting backdoors. The main reason is that it is very hard for attackers to tune its hyperparameters. The authors do not provide any details about how to choose parameters for this cyclical learning rate scheme. I also notice that the authors only evaluate ResNet models. Do we need different hyperparameters on different models?
- As mentioned in above point, the authors do not evaluate other pruning-based methods like ANP [1] and RNP [2]. I think the author need to evaluate these defense methods to show effectiveness of proposed attack method.
- Comparsions with existing advanced backdoor attacks: [3] proposed more stealthy and robust backdoor attacks without controlling training process. And, I also strongly suggest that the authors should evaluate more diverse attack settings like lower poisoning rate (1%).

2. Lacking of ture explanations: The intuition behind proposed method is too hauristic. Apart from evaluation of ASR, the authors do not provide any evaluation about proposed hypothesis. Actually, we are not sure whether it leads to more flat and stable minimum with inserted triggers. The author could provide loss landscape analysis to verify this point.
I think the authors are not familiar with loss landscape anaysis of DNN models. Adopting a cyclical learning rate scheme can not gaurantee searching a flat and stable local min. We could adopt SWA [4] to achieve this goal.

[1] Adversarial Neuron Pruning Purifies Backdoored Deep Models.
[2] Reconstructive Neuron Pruning for Backdoor Defense.
[3] Revisiting the Assumption of Latent Separability for Backdoor Defenses.
[4] Averaging Weights Leads to Wider Optima and Better Generalization.

**Questions:**

Please see Weaknesses.

---

> ### Author Response · Authors · 2023-11-23
> **Response to Reviewer EHtc**
>
> ### **1. The proposed attack method is specific to superFT**
> It is important to note that our training strategy is designed to counteract all finetuning-based defenses, despite being inspired by the idea of cyclical learning rate scheme used in superFT. As demonstrated in our paper, FMN effectively improves the attack robustness against a wide range of finetuning-based defenses, including vanilla finetuning, superFT, FT-SAM, and NAD.
> ### **2. It is not clear whether other defense methods are equally effective, such as ANP.**
> SuperFT is a quite strong defense, and in many cases it is more effective than other defenses such as ANP. One case study is with LIRA attack. As reported in Table 2 in our main paper, superFT can effectively mitigates the original LIRA attack. In constrast, as reported in Table 7 in our Appendix, ANP fails to mitigate this attack; it cannot reduce the ASRs without significant drops in BA.
> ### **3. How to choose parameters for this cyclical learning rate scheme? Experiments with different model architectures**
> We use the same hyper-parameters used in superFT for our cyclical learning rate scheme. We found this configuration is effective, and there is no need to tune these hyperparameters.
> We have run additional experiments utilizing various model architectures while maintaining the hyperparameters consistent with those employed for ResNet in our original study.
> We report the BA/ASR in the tables below. As shown, our method still achieves high ASR and remains effective against advanced fine-tuning defenses, such as super-FT and FT-SAM, across various architectures. We have included the results in the revised Appendix.
> | Model | Attack | No defense | Super-FT | FT-SAM
> |-------| -------- | -------- | -------- | --------
> | VGG16 | Blend + FMN  | 91.21/97.21  | 89.66/87.92  | 90.13/93.06
> | VGG16 |  Trojaning + FMN |  90.06/99.63 |  88.79/89.96 |  89.25/93.34
> | MobileNetv2 | Blend + FMN  | 93.73/98.99 | 91.44/93.29  | 91.62/95.82
> | MobileNetv2 | Trojaning + FMN | 93.57/99.97 | 90.59/93.54 |  90.94/96.07
> ### **4. Evaluation with pruning-based method**
> We indeed evaluated one instance of our attack, LIRA+FMN, against pruning-based defenses, including Fine-pruning and ANP, and reported the results in Section 4.6 of the main paper and Section A.6 of the Appendix. LIRA+FMN bypassed both defenses. Regarding the reviewer's recommendation, we have additionally conducted experiments with RNP and reported the results in the table below.
> | Attack | No defense | RNP |
> | ----- | ----- | ----- |
> | LIRA | 94.42/100 | 92.05/16.67 |
> | LIRA + FMN | 94.26/100 | 91.94/16.89 |
>
> It is important to note that our training strategy is specifically designed to enable existing attacks, such as LIRA, to counteract finetuning-based defenses. The robustness against other types of defenses will depend on the choice of the specific attack employed. As shown in the case study with LIRA+FMN, our method does not diminish the attack's effectiveness against other defense approaches.

---

> > ### Author Response · Authors · 2023-11-23
> > **Response to Reviewer EHtc (cont.)**
> >
> > ### **5. Comparison with advanced backdoor attacks like [3]**
> > Regarding the reviewer's suggestion, we have run experiments of [3] against advanced finetuning-based defenses. We report the experimental results of [3] and ours in the table below for comparison. As shown, [3]'s resilience against backdoor defenses is significantly lower compared to our method. We contend that [3] focuses on enhancing the latent separability between backdoor and clean samples, which primarily enhances attack performance against defenses that rely on latent separability assumptions but does not improve the resistant against advanced finetuning techniques.
> >
> > | Attack | No defense | Super-FT | FT-SAM |
> > | ------ | ------ | ------ | ------ |
> > | Adaptive-Blend ([3])| 92.42/100 | 90.14/29.96 | 91.43/32.19 |
> > | Adaptive-Patch ([3]) | 92.36/100 | 90.28/30.12 | 90.63/35.57 |
> > | Blend + FMN | 93.02/100 | 91.99/86.22 | 92.05/98.54 |
> > | Trojaning + FMN | 94.00/99.98 | 92.38/88.23 | 92.31/99.79 |
> >
> > ### **6. Evaluation with low poisoning-rate**
> > We thank the reviewer for the suggestion. We have run experiments with a low poisoning rate (1%) and report their results in the table below. With such a low poisoning rate, our attacks are less robust under Super-FT, but their ASRs are still above 50%. FT-SAM is weaker than Super-FT, allowing our ASRs to stay high, around 90%-100%.
> > | Attack | No defense | Super-FT | FT-SAM |
> > | ------ | ------ | ------ | ------ |
> > | Blend + FMN| 93.60/100 | 91.42/54.66 | 91.62/89.82 |
> > | Trojan WM + FMN | 93.57/99.97 | 90.64/65.73 | 91.55/98.90 |
> >
> > ### **7. Loss landscape analysis. Comparison to SWA.**
> > We thank the reviewer for the suggestion. We have added a loss landscape analysis to Section A.9 of the revised Appendix. The landscape for attack error rate from our method is much flatter than those from the original attack and its SWA version. Hence, our attack is resilient against finetuning-based defenses.
> > We also try replacing FMN with SWA. The results reported below suggesting that SWA is not robust against both Super-FT and FT-SAM, unlike our proposed technique:
> > | Attack | No defense | Super-FT | FT-SAM |
> > | ------ | ------ | ------ | ------ |
> > | Blend + SWA| 92.42/97.51 | 90.21/21.34 | 91.43/30.19 |
> > | Trojan WM + SWA | 92.81/98.69 | 89.94/25.12 | 90.26/34.52 |

---

### Official Review · Reviewer_Rq4g · 2023-10-29

**Soundness:** 3 good
**Presentation:** 3 good
**Contribution:** 2 fair
**Rating:** 6
**Confidence:** 4

**Summary:**

This paper presents FMN, a novel attack method to strength trojaning attacks (backdoor attacks) against DNNs. The key components of FMN are  the cyclical learning rate and the clean-backdoor interleaved training. The experimental results in this paper show that FMN successfully strengthens several existing trojaning attacks against several existing defense methods, i.e., the ASR remains high even if the backdoored model is purified by the defense methods.

**Strengths:**

1.FMN is compatible with various backdoor attacks.
2.The experimental results show that FMN can significantly increase the ASR of the mitigated backdoored model.

**Weaknesses:**

1.This paper should contain more analysis about the reason why  the cyclical learning rate and the clean-backdoor interleaved training work.
2.Though empowering existing backdoor attacks is an interesting idea, this paper should further investigate how to defend against FMN-powered backdoor attacks.

**Questions:**

1.What is the reason that the cyclical learning rate and the clean-backdoor interleaved training can work?
2.Are there any possible defense strategies against FMN?

---

> ### Author Response · Authors · 2023-11-23
> **Response to Reviewer Rq4g**
>
> ### **1. Analysis about why the cyclical learning rate and the clean-backdoor interleaved training work**
> Our experiments in ablation studies confirm the effectiveness of the combination of two training strategies: the cyclical learning rate and clean-backdoor interleaved training. Using only one strategy will lead to a low attack success rate. The cyclical learning rate helps to find flat areas, and the clean-backdoor interleaved training guarantees that it is the shared flat area of both function losses.
>
> We compare the flatness of losses of poisoned data and clean data sets between different methods to illustrate our arguments. Figure 6 in Appendix shows that the landspace of our loss function at the parameter solution is flatter than that of other methods.
>
> We also provide empirical intuition of why FMN is effective against finetuning-based defenses. Figure 7 in the revised Appendix shows the ASR and BA  along the connectivity path of the poisoned model and its corresponding fine-tuned version after undergoing the fine-tuning process of Super-FT defense. As can be observed, with conventional backdoor learning (left figure), when we linearly interpolate from the backdoored model to its corresponding Super-FT's fine-tuned version, the intermediate model's poisoning loss (i.e., the loss recorded only on the poisoned samples) increases, resulting in the decrease in ASR, while their clean losses and BAs are approximately stable. On the other hand, with FMN training (right figure), linearly interpolating between the backdoor and its corresponding Super-FT's fine-tuned model, the poisoning loss and ASR, as well as the clean loss and BA, are almost constant, indicating that FMN learns the backdoor in a region that makes it difficult for a fine-tuning defense to escape from.
>
> ### **2. Possible defense strategies against FMN**
> Our research highlights the potential risks of relying on third-party pre-trained models and underscores the importance of fostering trust between users and model providers. To defend against our attack, users should only use pre-trained models from trusted providers or actively participate in the training process. We also urge the research community to delve further into this area to develop more robust safeguards.
>
> We have included this discussion in the conclusion of the revised manuscript.

---

### Official Review · Reviewer_WyV7 · 2023-10-31

**Soundness:** 3 good
**Presentation:** 4 excellent
**Contribution:** 3 good
**Rating:** 6
**Confidence:** 5

**Summary:**

This paper explores a backdoor attack method called "Forget Me Not", which aims to overcome the catastrophic forgetting problem during model fine-tuning. Specifically, this paper proposes a backdoor learning method based on a cyclic learning rate policy that enhances the persistence of existing backdoor methods. The aforementioned method can bypass fine-tuning-based backdoor defenses and maintain effectiveness during complex fine-tuning processes. The authors demonstrate the effectiveness of the proposed method on three popular benchmark datasets.

**Strengths:**

1. Clear background introduction. the article provides readers with a thorough review of backdoor attacks and related work in the introductory section, providing good background knowledge.
2. Reasonable experimental design. The experimental setup and presentation of results in the article are clear, providing readers with an intuitive sense of the effectiveness of the method.
3. Clear charts and graphs. the charts and graphs are well-designed and help readers understand the content of the article.

**Weaknesses:**

1. Insufficiently detailed description of the methodology. When describing the "Forget Me Not" method, the details in some parts of the article are not clear enough. It is suggested that the authors provide more detailed algorithm description or pseudo-code in the method section so that readers can understand and reproduce better. 2.
2. The related work section can be expanded. Although the article has listed some related works, there are other important works in the field of backdoor attacks that can be referenced. In addition, I would like to know the rationale or justification for choosing these seven attack methods as representative methods. It is not possible for the authors to exhaust all the attack methods and prove the enhancement of backdoor persistence by the method, but I think representative methods need to be chosen to prove the comprehensiveness of the experiments.
3. Results of other defense experiments. Although the authors compare many fine-tuning-based defense methods to prove the effectiveness of the proposed backdoor, I am still concerned about whether the existing attack methods are able to overcome the existing backdoor defense methods, such as data cleansing methods, model modification methods, and model validation methods.
4. Analysis of defense strategies. Considering the potential threat of backdoor attacks, it is recommended that the authors discuss possible defense strategies or propose corresponding challenges in their articles.

**Questions:**

Please refer to the weaknesses.

---

> ### Author Response · Authors · 2023-11-23
> **Response to Reviewer WyV7**
>
> ### **1. More detailed description of the methodology**
> We thank the reviewer for the suggestion. We have added a detailed Algorithm block for FMN in the revised Appendix.
>
> ### **2. Rationale for choosing these seven attack methods**
> The chosen methods are representatives for different backdoor attack approaches:
> - BadNets and Blended: standard attacks with random trigger patterns. BadNets uses an image patch, while Blended uses a blended image as the trigger.
> - $L_p$ regularization and trojaning: optimized trigger patterns
> - Input-aware: image-dependent trigger
> - LIRA: imperceptible optimized trigger
> - Narcissus: optimized trigger, clean-label attack
>
> We have revised the manuscript to include the rationales.
>
> ### **3. Results of other defense experiments, such as data cleansing methods, model modification methods, and model validation methods**
> It is important to note that our training strategy is specifically designed to enable existing backdoor attacks to bypass finetuning-based defenses. Consequently, the robustness against other types of defenses will depend on the choice of attack used with our training process.
>
> As shown in the case study with LIRA+FMN in Section 4.6 of the main paper and Section A.6 of the Appendix, our method does not diminish the attack's effectiveness against other defense approaches. LIRA+FMN is robust against various defense mechanisms, including Neural Cleanse, Fine-pruning, ANP, and STRIP. Note that we test only representative backdoor defenses that LIRA has already bypassed. Also, we do not include data cleansing defense from our evaluation because our threat model assumes that the adversary provides the backdoored model to the user for deployment rather than supplying malicious data.
>
> ### **4. Discussion of possible defense strategies**
> We thank the reviewer for the suggestion. Our research highlights the potential risks of relying on third-party pre-trained models and underscores the importance of fostering trust between users and model providers. To defend against our attack, users should only use pre-trained models from trusted providers or actively participate in the training process. We also urge the research community to delve further into this area to develop more robust safeguards.
>
> We have included this discussion in the conclusion of the revised manuscript.

---

### Official Review · Reviewer_xTDh · 2023-11-05

**Soundness:** 2 fair
**Presentation:** 3 good
**Contribution:** 2 fair
**Rating:** 5
**Confidence:** 4

**Summary:**

The key contribution of this paper is to demonstrate that fine-tuning (regardless of the techniques used) is not a proper defense against backdoor attacks. The paper specifically evaluates the two fine-tuning approaches: super fine-tuning and FT-SAM by Zhu et al. By simulating their fine-tuning mechanisms in backdoor training, the attacker can inject backdoors more robust to the fine-tuning techniques. In evaluation with various backdoor attacks, the paper shows their backdoor attacks become more resilient against fine-tuning.

**Strengths:**

1. The paper shows (advanced) fine-tuning cannot be a backdoor defense.
2. The paper runs extensive experiments to validate their claims.
3. A well-written paper; easy to follow.

**Weaknesses:**

1. Unfortunately, we don't believe fine-tuning can be a backdoor defense.
2. The paper is written to prove the point that we already believe.
3. The experimental results are not 100% convincing.

Detailed comments:

I like this paper showing (or re-confirming) that fine-tuning cannot be an effective defense against backdoor attacks. Even if there are manuscripts making bold claims like "fine-tuning is effective," I don't believe that it is the case: their positive results are coming either (1) from running fine-tuning with longer epochs or large learning rates (often) or (2) from an adversary unaware of their fine-tuning methods.

So, I am a bit positive to have this paper as empirical evidence showing that existing claims are not based on a concrete security analysis.

----

However, we also know that fine-tuning cannot be a defense; a vast literature on backdoor attacks evaluated fine-tuning and confirmed that it is ineffective (note that it is not against this advanced fine-tuning). We already have a skepticism about fine-tuning.

So, on the other hand, it is less scientifically interesting to prove that we already know with empirical evaluation. I am a bit confident that even if the two prior works are out to the community, no one will believe that fine-tuning can become an effective countermeasure.

----

I also find that the paper puts a lot of effort into emphasizing fine-tuning as a defense seriously considered in the community. But it often gives an incorrect view of the prior work, which I want the authors fixing them before this manuscript will be out to the community.

For example, papers like NeuralCleanse do not consider fine-tuning as a primary mean to defeat backdoors. The key idea was to "reconstruct" the trigger from a set of assumptions about backdooring adversaries. Once we know what was used as a backdoor trigger, the fine-tuning is a natural next step to "unlearn" the backdoor behaviors. It is not the same as one uses fine-tuning without knowing the trigger, which should be addressed and fixed in the paper.

I found more like this in the introduction and backdoor defense section.

----

Finally, sometimes fine-tuning reduces their attack's success rate. This (as-is) can be shown as the effectiveness of fine-tuning (as at least the success rate has been decreased).

To be a more concrete claim, the results have to be compared with a baseline. What would be the baseline? The cases where we reduce the attack success rate to 50%? It was not clear in the paper; therefore, the claims discussing the effectiveness of fine-tuning can also be controversial ---even if I don't believe that fine-tuning works.

----

At the moment, I am slightly leaning toward rejecting this paper. But if those concerns are (and will also be) addressed in the responses, I will be happy to bump up my assessment.

**Questions:**

My questions are in the detailed comments above.

**Details Of Ethics Concerns:**

No concern about the ethics.

---

> ### Author Response · Authors · 2023-11-23
> **Response to Reviewer xTDh**
>
> ### **1. Regarding concerns about finetuning-based defense without knowing the trigger**
> We acknowledge that the vanilla fine-tuning (FT) has limited capability in mitigating backdoors. However, it still offers some level of protection and is often combined with other components to establish more robust defenses. FT-based defense remains an active area of research, with methods like NAD, FT-SAM, and Super-FT demonstrating empirical effectiveness. Particularly, Super-FT was released at the end of 2022, while FT-SAM has been published at ICCV 2023. In its paper, FT-SAM  showed stronger defense performance than popular defenses in other approaches, including Neural Cleanse, ANP, ABL, and iBAU. In our paper, both Super-FT and FT-SAM can effectively mitigate LIRA, while FP and ANP fail (as reported in Section A.6 in the Appendix). Finally, as shown in our response to Reviewer EHtc, both Super-FT and FT-SAM can effectively mitigate very recent advanced attacks proposed in [1]. Hence, advanced finetuning-based backdoor defenses are serious mechanisms to protect users from the backdoor threat. Our method aims to enhance the robustness of backdoor attacks against these advanced FT defenses.
>
> Also, we understand the reviewer's concern, and we have refined the discussion of fine-tuning-based defense in the revised introduction to provide a more accurate overview of this defense approach.
>
> [1] Revisiting the Assumption of Latent Separability for Backdoor Defenses. In *ICLR* 2022
>
> ### **2. Regarding finetuning-based defense after knowing the trigger**
> Thanks for your useful suggestion. Our paper aims to address the finetuning-based defenses without knowledge of the trigger, which is an active area of research, with methods like NAD, FT-SAM, and Super-FT. Hence, we do not focus on the application of finetuning in mitigating the backdoor after knowing the trigger, whose effectiveness relies much on the trigger reconstruction rather than the finetuning itself. We have added a clarification in our revised paper.
>
> ### **3. Baseline for evaluation**
> We chose 50% ASR as the threshold to determine whether an attack is successful. This level of accuracy indicates that backdoor samples are more likely to be misclassified as the target label. We have elaborated on this point in the revised manuscript.

---

### Author Response · Authors · 2023-11-23
**Response to all reviewers**

Our work proposes FMN, a novel backdoor training approach that can enable existing backdoor attack techniques to bypass recently proposed advanced finetuning backdoor defenses. Despite being inspired by superFT, FMN's training can help an attack bypass a wide range of finetuning defenses, including vanilla finetuning, superFT, FT-SAM, and NAD. We have also confirmed that FMN does not reduce the utility of existing backdoor attacks; i.e., if the existing attack (e.g., LIRA) previously bypassed a defense (e.g., Fine-pruning), that attack when training with FMN will still be able to bypass this defense, or FMN only make the attack even more powerful. This is a significant threat to securing ML models against backdoor attacks, which urges the model users to establish trust in the model acquisition process and urges the backdoor researchers to develop more robust safeguards against this type of defense.

During the rebuttal process, we have:
* Provided evidences showing that advanced finetuning-based defenses are serious mechanism against backdoor attacks.
* Provided additional experiment details, including detailed algorithm, rationale for attack choices, and the baseline for evaluation.
* Provided additional analysis of why FMN works, especially the analysis of the loss landscapes.
* Provided our perspective on possible defense strategy against FMN.
* Provided experiments with different model architectures, showing that our FMN remains effective without tuning the hyper-parameters.
* Clarify and provide extra experiments for pruning-based defenses. Note that since we mentioned that FMN does not reduce the utility of an existing attack, we focus our attention on the evaluation of finetuning-based defenses.
* Provided experiments with low poisoning rates. FMN is still effective.
* Provided additional comparisons with advanced backdoor attacks and demonstrated that FMN is a much greater threat for finetuning defenses.
* Provided references to sections in our paper to answer the reviewers' questions.

We have added these discussions in the revised submissions. We thank the reviewers for all the constructive suggestions and we hope that our responses have addressed the reviewers' concerns. FMN is a real and sophisticated threat, especially when combined with the existing attacks, and we encourage the security community to develop countermeasures against FMN. Finally, we are happy to answer additional questions promptly.

---

### Meta-Review · Area_Chair_Fban · 2023-12-10

**Metareview:**

This work proposed a backdoor attack methods with stronger resistance  to fine-tuning based defense. It proposed a novel training algorithm with a cyclical learning rate scheme and Clean-backdoor interleaved training procedure.

Most reviewers recognized the effectiveness of enhancing several existing backdoor attacks against fine-tuning based defenses. However, the main concerns covers the theoretical analysis about the mechanism of the defense method, the effectiveness against other types of backdoor defenses, as well as the adaptive defense. The authors didn’t provide very satisfied responses during the rebuttal.

I appreciate the value of this work’s goal, and encourage the authors to further explore the behind mechanism.

**Justification For Why Not Higher Score:**

see above

**Justification For Why Not Lower Score:**

n/a

---

### Decision · Program_Chairs · 2024-01-16

Reject